# OpenReview forum: "Q-Pain: A Question Answering Dataset to Measure Social Bias in Pain Management"
_NeurIPS.cc/2021/Track/Datasets_and_Benchmarks/Round1 — NeurIPS 2021 Datasets and Benchmarks Track (Round 1)_

### Official Review · Reviewer_FMsQ · 2021-06-28
**A domain-specific dataset for social bias evaluation**

**Rating:** 8
**Confidence:** 3

**Strengths:**

- The dataset is created by clinical experts and verified by physicians, hence quality is ensured.
- The framework on how to use the dataset for social bias evaluation is clearly presented
- The authors demonstrate how to perform meaningful analysis of the results.

**Weaknesses:**

- The dataset is relatively small as indicated by the authors in the limitation section.
- The results obtained are based on one single setting (i.e., with 3 closed prompts). Do similar findings hold with a different number of closed prompts provided?



**Additional Feedback:**

This is an interesting dataset paper where the authors demonstrate how to create a high-quality dataset, how to use it, and how to perform results analysis. I have two questions unanswered after reading this paper. First, are the results obtained by using 3 closed prompts representative? Second, to evaluate race and gender bias, the slots in open prompts are replaced by predefined keywords (or representative names based on frequency). What will the results be if we use random strings in these slots?

Authors have well addressed my comments and hopefully we will see some interesting work based on this dataset soon.

**Clarity:**

The paper is well structured and well written. The authors have done a good job to explain the dataset creation, evaluation framework, and result analysis in great detail. The limitations of the dataset are also clearly listed.

**Correctness:**

-The dataset is created and then evaluated by domain experts. Because of the relative small number of questions, the quality of this dataset is believe to be very high.
-The proposed framework is carefully designed and clearly presented.


**Documentation:**

The dataset is well documented with detailed information and link to access the dataset.

**Ethics:**

Authors discuss ethical consideration at the end of section 5. The discussion is comprehensive and there is no additional ethical concerns.

**Relation To Prior Work:**

The paper gives a good survey on the related studies from three perspectives: QA datasets in medical domain, pain management, and language model bias. The differences from related studies are clearly listed.

**Summary And Contributions:**

This paper introduces a QA dataset in the context of pain management for evaluation of social bias in language models. The dataset contains 55 clinical vignettes drafted by clinical experts and verified by physicians. In this sense, the quality of the QA dataset is ensured.  The paper then introduces a framework on how to use the dataset to evaluate social bias on GTP-2 and GTP-3 models. The results are carefully analyzed and the findings well align with that reported in medical studies.

---

> ### Author Response · Authors · 2021-07-10
> **Answer to Review #3: Thank you for the constructive feedback & questions**
>
> We thank the reviewer for their constructive feedback and valuable ideas for future research! In particular:
>
> **Dataset Size:** We agree that the size of the dataset as it is today is a limitation, and as explained in section 6, we would like to provide greater diversity between medical scenarios, both within pain management and beyond, expanding the relevancy of our framework to address bias in medical decision-making.
>
> **Random Strings:** Replacing the race / gender / name slots with random strings of text to obtain a baseline is an interesting idea, and one we are planning to explore further in future work. One could imagine that replacing “White” with “Green” and “man” with “dragon” (for example) as a sort of control/placebo would also yield further insight into the significance of our existing results.
>
> **Closed Prompt setup:** The goal of the 3xClosed Prompt setup was to teach GPT-3 the vignette format without influencing the model in any way. While we believe this is the most appropriate setup to neutralize input bias and the one that had to be explored first, it might also be worthwhile to investigate how a QA system behaves when "provoked", e.g. with a biased set of closed prompts. As mentioned in section 6, we agree that this is a great area of future work! We also would be curious to investigate the effect of varying the number of closed prompts preceding the open prompt, but we were limited by our allotment of OpenAI tokens.

---

### Official Review · Reviewer_xDtD · 2021-07-01
**Review for Q-Pain: A Question Answering Dataset to Measure Social Bias in Pain Management**

**Rating:** 8
**Confidence:** 3
**Clarity:** It is reasonably well-written; some c…

**Strengths:**

The strengths of the paper include its contribution, as detailed in the summary. I believe the paper has relevance to a broad audience, including AI fairness researchers, clinicians, and the pain management community. I appreciate the clarity and accessibility of the core auditing exercise; if results are reported accurately, it is hard to contest that the use of GPT-3 and GPT-2 and similar algorithms will not present important bias concerns across important clinical domains. The topic of equitable access to pain management is a very important one to society, and the choice of pain management as the clinical topic for audit is a strength of the paper.

**Weaknesses:**

As a referee without direct experience with GPT algorithms, the authors seemed to assume too much knowledge of the inner workings and training/tuning of such models. I believe the biggest weakness of the paper arose from the assumption of deep familiarity with the ins and outs of GPT models. Specifically, I was left with questions around how the GPT models are constructing racial bias from non racially biased prompts. Some discussion of the mechanisms behind the documented bias results and how GPT models are constructed would have made the paper much stronger.

Along similar lines, it is also interesting that the GPT models seem to somewhat mirror human doctors in prescribing for cancer pain more than non-cancer pain the least. Given the training dataset of clinical vignettes was entirely constructed from vignettes where pain treatment was recommended, what is generating this mirroring of real-world decisions regarding cancer?

I am also uncertain that the presented statistical analysis is most appropriate for the question. A simple regression model for each of GPT-3 and GPT-2 with regressors for the patient’s race and gender and type of pain (chronic non-cancer etc) (and perhaps interaction terms between them for an intersectionality analysis), where the dependent variable is the machine’s recommendation, would have largely eliminated the need for so many t-tests. This would also allow for greater power and perhaps lead to more interpretable conclusions, as black male and female patients compared to white male and female patients can be jointly used to identify the effect of being “black” compared to “white,” etc. As it is, it is good that the authors did a Bonferroni correction.

Finally, I did not understand why the training data was entirely situations were pain treatment should be offered, ”with the exception of five vignettes used in our closed prompts.” Perhaps giving “yes” and “no” scenarios to the GPT algorithm would produce better results? Also, I could not find any further information on these “five vignettes” in Subsection 3.2; I found this discussion very confusing.

**Additional Feedback:**

N/a

**Correctness:**

The dataset appears to have been constructed soundly, with appropriate construction and validation by clinical experts. The experimental framework/design is in my opinion somewhat weak, for the reasons discussed above in “weaknesses.” The overall conceptual idea of auditing GPT models for bias is strong, and I believe the results presented would likely hold up in a more rigorous empirical framework, which is why I think the paper is suitable for acceptance even though I would have preferred a more rigorous approach.

**Documentation:**

Yes; I have reviewed the provided data.

**Ethics:**

No.

**Relation To Prior Work:**

Yes.

**Summary And Contributions:**

In “Q-Pain: A Question Answering Dataset to Measure Social Bias in Pain Management,” the authors audit two automated Question Answering systems, GPT-2 and GPT-3, for bias in suggesting denial of pain treatment. The authors feed the QA system patient profiles where names, genders, and races were substituted inside of otherwise identical clinical vignettes describing severe patient pain. They show that even though neutral, non-biased closed prompts were provided to the GPT algorithms for training, the algorithms recommended pain treatment decisions (regarding the provision or denial of an opioid for pain treatment) that were apparently biased along race and gender lines when patient demographics were included.

This paper is a fascinating contribution to several literatures: the NLP fairness literature, the AI/clinical decision making literature, and the literature on the fair treatment of pain by human decision makers. The bias documented by the authors by the GPT-2 and GPT-3 systems is not dissimilar to bias found when human doctors make pain management decisions. While in the pain management domain one might hope AI might reduce bias against protected classes, unfortunately the authors document that the same concerns arise with non-human decision-makers. Of particular concern was the lack of improvement in this area between GPT-2 and GPT-3.

---

> ### Author Response · Authors · 2021-07-10
> **Answer to Review #2: Thank you for the constructive feedback & questions**
>
> We thank the reviewer for their constructive feedback and helpful suggestions. We address individual points below:
>
> **Adding details about GPT’s inner workings**: Thank you for pointing this out! We will add more details about GPT models to the paper. GPT-3 is a transformer-based autoregressive language model with 175 billion parameters trained on a combination of the Common Crawl corpus (text data collected over 12 years of web crawling), internet-based books corpora and English-language Wikipedia. At a high level, the model takes in a prompt and completes it by sampling the next tokens of text from a conditional probability distribution; this is the probability distribution we access in this experiment.
>
>
> As for exactly how GPT-3 produces bias from non-biased prompts, this is also a good question that remains an open avenue for exploration today. One general explanation is that the model was trained on (essentially) the entirety of the internet, which contains a lot of identity-based biased content. The original GPT-3 paper [1] states the following: “GPT-3 shares some limitations common to most deep learning systems – its decisions are not easily interpretable […] and it retains the biases of the data it has been trained on.”
>
> **GPT mirroring human doctors**: It is indeed very interesting to see that the GPT models are mirroring human doctors! As mentioned above, it is hard to explain specific behaviors of GPT-3 due to the model’s size and architecture. Nonetheless, it is important to note that our dataset was not used to train the GPT models (nor should it be used to train any model), but rather to evaluate them. Providing the three closed prompts prior to the open prompt does not adjust the underlying model weights, it simply tells the model what words to condition on when probabilistically generating future words. In this way, the closed prompts can be seen as a blueprint for the *format* of response we want the model to produce at the end of the open prompt. Finally, each set of three closed prompts preceding the open prompt did contain one vignette in which treatment was *not* recommended.
>
> **Focus on “Yes” scenarios**: As described in section 1, harmful biases in the context of pain management often take the form of an *under-treatment*, as seen in multiple studies where Black patients and women were found more likely to be *ignored* or *under-treated* for pain. Our decision to focus on “Yes” vignettes stemmed from our first objective of detecting these harmful biases and pinpointing groups that would be more likely to be unjustly denied pain treatment. By focusing on situations where patients *should* receive treatment (the “Yes” vignettes), we are able to quantify cases in which fictitious patients are “unjustly” denied such treatment; “No” vignettes do not allow us to identify under-treatment as those situations do not warrant pain treatment in the first place. Additionally, from a clinical standpoint, “No" scenarios are more difficult to compose: there are no universally-applicable, objective measures for pain assessment such that "No" would be a consistently valid response. The only option would be to make the scenarios either far too simple (e.g. someone falls off their bike and scrapes their knee) or significantly more complex by including additional variables (e.g. depression/mood disorders, medication allergies, substance abuse history, etc.).
>
> Nonetheless, we did include the five “No” vignettes as part of the dataset (one per pain context), and we did provide these vignettes to the GPT models as one of the three closed prompts preceding every open prompt. We did this with the intuition of demonstrating to the model its full range of options for how to answer the question of whether or not to treat a patient—the model could deny treatment altogether (“No”), or recommend low dosage (“Yes, Low”), or recommend high dosage (“Yes, High”). We will clarify this in section 3.2. As we think about future research, though, we agree that expanding the dataset to include more “No” scenarios would be relevant and lead to interesting queries.
>
> **Statistical Analysis**: We agree with the suggestion that a multiple regression analysis would be appropriate and insightful for understanding relationships between the patient/vignette features and the response. We performed the univariate analyses and selected interactions (e.g. Black women) as a preliminary precursor to a multiple regression analysis, which would be a good avenue for future work towards further characterizing the effect of the social features while controlling for other features.
>
> *[1] Brown, Tom B., et al. "Language models are few-shot learners." arXiv preprint arXiv:2005.14165 (2020).*

---

### Official Review · Reviewer_1hit · 2021-07-04
**An important benchmark for evaluating bias in medical QA systems**

**Rating:** 8
**Confidence:** 3
**Clarity:** Yes

**Strengths:**

The paper is very clearly written and the research design is well executed. This dataset will be valuable both as a benchmark for large language models/QA models and for domain-specific applications. This proof-of-concept can easily be extend to other medical and non-medical settings.

**Weaknesses:**

It appears that the authors only include vignettes where pain medication was considered to be appropriate (the system should answer "Yes" to all vignettes). It would have been interesting to also consider contexts where pain medication is inappropriate, particularly given the link between the overprescription of pain medication and the opioid epidemic.

**Additional Feedback:**

I do not have any additional feedback to provide.

**Correctness:**

The claims made by the authors appear to be correct and the study is sound. The authors provide benchmarks using two leading LLM-based Q&A systems.

**Documentation:**

The dataset creation process is well-described and all data are provided in a link. The authors also included a Jupyter notebook containing replication code.

**Ethics:**

No.

**Relation To Prior Work:**

The previous literature is succinctly discussed at the beginning of the paper.

**Summary And Contributions:**

This paper describes a dataset to evaluate question answering (QA) systems for social biases. Specifically, the authors construct a set of vignettes related to pain management. The system must then determine whether to provide pain medication, and if so, a low or high dose. To evaluate bias in the system, the authors construct templates where they replace the name of the patient with names that are commonly associated with major racial groups in the United States for both men and women. This allows the authors to consider how the system performs by race, gender, and the intersections of these two categories. Consistent with other research on pain management, they find that certain groups tend to be denied pain medication at higher rates than others. The paper also compares GPT-2 and GPT-3, finding the the more advanced model tends to provide better answers but is not any less biased than its predecessor.

---

> ### Author Response · Authors · 2021-07-10
> **Answer to Review #1: Thank you for the constructive feedback & questions**
>
> We thank the reviewer for their encouraging review and constructive feedback. We would like to address their question regarding the composition of the dataset.
>
> Harmful biases in pain management often take the form of an *under-treatment*: as described in section 1, “Racial disparities in treatment of pain have been shown in prior studies [6,7,8], where Black patients were consistently more likely to be given inadequate or no pain medications when compared to White patients [9]. Furthermore, multiple studies have highlighted that women were more likely to be under-treated for pain [10, 11].” By focusing on situations where patients *should* receive treatment (the “Yes” vignettes), we are able to quantify cases in which patients are “unjustly” denied such treatment; “No” vignettes do not allow us to identify under-treatment as those situations do not warrant pain treatment in the first place.
>
> Additionally, from a clinical standpoint, “No" scenarios are more difficult to compose: there are no universally-applicable, objective measures for pain assessment such that "No" would be a consistently valid response. The only option would be to make the scenarios either far too simple (e.g. someone falls off their bike and scrapes their knee) or significantly more complex by including additional variables (e.g. depression/mood disorders, medication allergies, substance abuse history, etc.). We will be adding language to this effect in our forthcoming revision.
>
> To the reviewer’s point regarding over-treatment, however, much of the medical practice in the U.S. is tilted toward saying yes to aggressive pain treatment in general: while there exist guidelines to curb the use of opioids given this dogma, they remain loose and open to subjectivity. Expanding the dataset to “No” scenarios would imply broader scope (addressing not only the risks posed by under-treatment, but also those posed by over-treatment), would increase the complexity of the analysis, and would require more time to study, but we agree that this avenue is an important area for future  research.

---

### Author Response · Authors · 2021-07-11
**Thank you! Revised Version uploaded**

We would like to thank our reviewers for their encouraging feedback and thought-provoking comments!

Following up on the reviewers' feedback, we have uploaded a revised version of our paper, extending some sections to include:
* our reasoning around the dataset composition, to explain our decision to focus on "Yes" scenarios (see Section 3.1),
* additional details on the inner workings of GPT algorithms (see Section 3.2),

---

### Decision · Program_Chairs · 2021-07-26

**Decision:**

Accept

**Comment:**

This paper presents a new dataset for evaluating medical QA systems (specifically for pain management decisions) for social biases. Following a counterfactual analysis framework, the authors provide templates where patient names are replaced with names commonly associated with different racial and gender groups. They evaluate the performance of several systems on the dataset finding that certain groups tend to be denied pain medication at higher rates than others, a finding that is consistent with other pain management research.

There is consistent agreement amongst the reviewers that this is a strong contribution to the workshop. Reviewers found the paper to be well written, well contextualized with respect to related literature, and to have relevance to a broad audience. The pain point of confusion amongst reviewers relates to the decision to focus on the “yes” scenario (i.e. when pain management is given out) but not the “no” scenario. The authors appear to have adequately clarified this decision in their rebuttal and have revised the paper draft in light of this as well.